# A Systematic Review and Appraisal of Epidemiological Studies on Household Fuel Use and Its Health Effects Using Demographic and Health Surveys

**DOI:** 10.3390/ijerph18041411

**Published:** 2021-02-03

**Authors:** Daniel B. Odo, Ian A. Yang, Luke D. Knibbs

**Affiliations:** 1School of Public Health, The University of Queensland, Herston, QLD 4006, Australia; l.knibbs@uq.edu.au; 2College of Health Sciences, Arsi University, Oromia, Asella P.O. Box 193, Ethiopia; 3Thoracic Program, The Prince Charles Hospital, Metro North Hospital and Health Service, Chermside, QLD 4032, Australia; i.yang@uq.edu.au; 4UQ Thoracic Research Centre, Faculty of Medicine, The University of Queensland, Brisbane QLD 4032, Australia

**Keywords:** cooking fuel, household air pollution, health effects, Demographic and Health Survey, DHS, low- and middle-income countries

## Abstract

The domestic combustion of polluting fuels is associated with an estimated 3 million premature deaths each year and contributes to climate change. In many low- and middle-income countries (LMICs), valid and representative estimates of people exposed to household air pollution (HAP) are scarce. The Demographic and Health Survey (DHS) is an important and consistent source of data on household fuel use for cooking and has facilitated studies of health effects. However, the body of research based on DHS data has not been systematically identified, nor its strengths and limitations critically assessed as a whole. We aimed to systematically review epidemiological studies using DHS data that considered cooking fuel type as the main exposure, including the assessment of the extent and key drivers of bias. Following PRISMA guidelines, we searched PubMed, Web of Science, Scopus and the DHS publication portal. We assessed the quality and risk of bias (RoB) of studies using a novel tool. Of 2748 records remaining after removing duplicates, 63 were read in full. A total of 45 out of 63 studies were included in our review, spanning 11 different health outcomes and representing 50 unique analyses. In total, 41 of 45 (91%) studies analysed health outcomes in children <5 years of age, including respiratory infections (*n* = 17), death (all-cause) (*n* = 14), low birthweight (*n* = 5), stunting and anaemia (*n* = 5). Inconsistencies were observed between studies in how cooking fuels were classified into relatively high- and low-polluting. Overall, 36/50 (80%) studies reported statistically significant adverse associations between polluting fuels and health outcomes. In total, 18/50 (36%) of the analyses were scored as having moderate RoB, while 16/50 (32%) analyses were scored as having serious or critical RoB. Although HAP exposure assessment is not the main focus of the DHS, it is the main, often only, source of information in many LMICs. An appreciable proportion of studies using it to analyse the association between cooking fuel use and health have potential for high RoB, mostly related to confounder control, exposure assessment and misclassification, and outcome ascertainment. Based on our findings, we provide some suggestions for ways in which revising the information collected by the DHS could make it even more amenable to studies of household fuel use and health, and reduce the RoB, without being onerous to collect and analyse.

## 1. Introduction

The World Health Organization (WHO), in its indoor air quality guidelines, defines solid fuels, including coal and biomass (e.g., charcoal, wood, dung and crop residues), and kerosene as "polluting". Households that use these fuels are exposed to household air pollution (HAP) [1]. Despite progress towards universal access to clean household energy, 47% of global households (~3.6 billion people) continue to depend on polluting fuels [2]. Combustion of these fuels in the household, for cooking, heating and lighting, emits particulate and gaseous pollutants that harm human health, contribute to increased ambient air pollution, and affect climate change [1,3]. It is estimated that women and children incur the greatest health burden, due to spending more time in and around the home in low- and middle-income countries (LMICs) [4,5].

Exposure to air pollution is the fourth-leading risk factor for disease burden worldwide, accounting for nearly 7 million premature deaths and more than 213 million disability-adjusted life-years (DALYs) in 2019 [6]. Each year, an estimated 3 million premature deaths and 91 million healthy years of life are lost due to illness attributable to HAP alone [6]. These HAP-related deaths and DALYs are nearly two times higher in countries with low socio-economic status [7]. HAP from cooking with polluting fuels is associated with many adverse health outcomes, of which the highest level of evidence is for cardiovascular and respiratory diseases, diabetes, cataracts, low birthweight and short gestation (preterm birth) [1,6,8].

Broad-scale evidence on the health burden of exposure to HAP in LMICs comes mainly from national surveys because the ideal methods, such as biomarkers, personal and micro-environmental monitoring are not feasible in those settings. The Demographic and Health Survey (DHS), which is funded primarily by the United States Agency for International Development (USAID), is a nationally representative household survey that has been used to collect data on population, health and nutrition in more than 90 LMICs since 1984 [9]. It is the main, often only, source of essential data in LMICs. DHS data have been used to calculate more than 30 indicators, supporting tracking the Sustainable Development Goals (SDGs) [10,11]. DHS data are crucial in a wide variety of research activities and policy decisions, including the allocation of health resources [12].

Globally, estimates of the proportion of the population using polluting fuels [1], and the resultant health risks in LMICs, is based on data collected primarily by DHS surveys [6,13]. In those countries, exposure to HAP is assessed indirectly using the type of fuel used for cooking as a proxy [14]. Numerous studies have reported associations between HAP and adverse health effects by performing secondary analyses of DHS data. The DHS surveys are periodically revised by soliciting feedback from end users with content expertise. Several extensive reviews and meta-analyses of HAP and health effects included DHS-based studies along with other relevant non-DHS studies [15,16,17,18,19]. However, to our knowledge, no reviews have focused solely on DHS-based epidemiological studies of HAP.

Here, we aimed to: (i) identify and collate all relevant peer-reviewed epidemiological studies of DHS-derived HAP estimates and health performed globally, (ii) determine what variables and analytical approaches can place studies at greater risk of bias, using a novel rating tool, and (iii) identify additional variables that could boost utility without being onerous to collect in resource-limited settings, given that the main purpose is not to collect data for HAP research. Due to practical constraints, this review did not include articles published in languages other than English.

## 2. Materials and Methods

### 2.1. Literature Search

The systematic review followed the approach detailed in the Preferred Reporting Items for Systematic Reviews and Meta-Analyses (PRISMA) guidelines [20]. Three electronic databases, including PubMed, Web of Science, and Scopus were used to search peer-reviewed articles. The DHS program publication portal and the reference lists of the included primary studies and review articles were searched for additional studies.

Medical Subject Headings (MeSH), free-text terms and keywords used in the search include air pollution (household air pollution, indoor air pollution and indoor), survey characteristics (family characteristics, household characteristics, health surveys and cross-sectional studies) and cooking activities and fuel types (cooking, indoor cooking, outdoor cooking, kitchen, fuel, cooking fuel, heating fuel, wood, kerosene, charcoal, coal, biomass and solid fuel). These were combined with Boolean operators “AND/OR” and the search strategy was customised to each database (Appendix A). The search strategy was developed in consultation with a librarian experienced in systematic reviews of epidemiological studies. The initial search was undertaken in June 2019, and an updated search, restricted to articles published after that date, was performed in September 2020. The updated search was done for PubMed only.

### 2.2. Review Protocol and Inclusion Criteria

The detailed methods of the analysis and inclusion criteria were pre-specified and registered in the PROSPERO international prospective register of systematic reviews (registration number: CRD42019137937) [21]. In brief, peer-reviewed studies that were based entirely on DHS data and considered HAP as a main exposure variable were included, with no restriction on the type of health outcome, geographic location (country), age or gender of study participants. All searches were restricted to articles published in English since the inception of the DHS program (1985) through to September 2020.

The search results from the main databases and the DHS publication portal were imported into EndNote X9. Duplicate records were then removed. The first reviewer (D.B.O.) screened the title and abstract of the remaining records against the inclusion criteria. A second reviewer (L.D.K.) independently reviewed titles and abstracts of a randomly selected subset comprising 10% of the overall records (i.e., after removal of duplicates). If disagreement occurred, it was resolved by further discussion to reach a consensus regarding inclusion or exclusion. In the course of data abstraction, authors of two articles were consulted for detailed clarification.

### 2.3. Risk of Bias and Quality Assessment

The quality and risk of bias (RoB) of studies meeting the inclusion criteria were assessed, using a new RoB instrument for non-randomized studies (NRS) of exposures, [22] by one of the authors (D.B.O.). This tool seeks to assess the methodological quality of the evidence and the RoB of studies of environmental exposures. It compares the quality and RoB of each study with a hypothetical randomised target experiment, rather than a study-design directed quality and RoB assessment approach [22]. The three steps involved in applying this instrument were: (1) present the review question, potential confounders, and exposure and outcome measurement accuracy information, (2) describe each eligible study as a hypothetical target experiment and including specific confounders from that study that will require consideration, and (3) assess RoB across seven items on the strengths and limitations of studies. Detailed guidance on the application of the tool is available elsewhere [22], including how to interpret and present the RoB of the studies, and using that information to make a transparent judgment.

The seven domains of RoB items used include bias due to confounding, bias in the selection of participants, bias in the classification of exposure, bias due to the departure from intended exposure, bias due to missing data, bias in the measurement of outcome, and bias in the selection of reported results. The final judgment for each RoB item was rated as low, moderate, serious or critical [22] (Appendix A). As a general indicator, a "low" RoB study would control important confounders listed in the RoB assessment table, would ascertain health outcome objectively (with validated method) and supported with record linkage (Appendix A). Such studies that scored higher RoB, particularly studies that were labelled under critical RoB were expected to be dropped. However, because our aim was to identify and collate all DHS-based studies on household fuels and health, including the benefits and weaknesses of using DHS data, we did not exclude any studies based on RoB scores. We instead used them to highlight some of the important considerations and challenges of using DHS data and various analytical methods used by the different studies.

We conducted the RoB assessment at the outcome level. Because RoB can vary within a study if multiple outcomes are assessed. We, therefore, evaluated RoB for each outcome.

## 3. Results

### 3.1. Search Results and Screening

The PRISMA flow chart is shown in Figure 1. The main database search returned 3396 records, plus 21 studies obtained from the DHS publication search portal. Of the 2748 records left after the removal of duplicates, 63 full-text records were downloaded for further assessment. The remaining 2685 records were excluded because they were not relevant to our review (see Figure 1 for details). Of the full-text records reviewed, 45 (71%) studies met our inclusion criteria (Figure 1). Table 1, Table 2, Table 3, Table 4, Table 5 and Table 6 contain a detailed summary of the included studies, including year, country, health outcomes, statistical methods, sample size, among others.

### 3.2. Description of Included Studies

Overall, data from 57 countries were included in these 45 studies. Three of 45 studies assessed multiple health outcomes [23,24,25], while the remainder assessed a single outcome, for a total of 50 unique analyses. All studies were cross-sectional in design. The country most frequently analysed was India (13/45 studies) [25,26,27,28,29,30,31,32,33,34,35,36,37], followed by Nepal (4 studies) [38,39,40,41], then Bangladesh [23,24,42], Nigeria [43,44,45] and Pakistan [46,47,48] (3 studies each). South Africa [49,50] and Zimbabwe [51,52] each accounted for two studies, respectively. The remaining studies focused on Ghana [53], the Philippines [54], Tanzania [55], Swaziland [56], Uganda [57], Afghanistan [58] and Malawi [59].

For multi-country analyses, a study of child mortality included 47 countries [60]. Three studies [61,62,63] combined multiple data from different sub-Saharan African countries and the other three studies pooled data from countries in different regions [64,65,66].

The included studies were published between 1999 and 2020, with the majority covering the period 2013–2020. The DHS data used by these studies were collected between 1992 and 2018. ARI in children is the most frequently reported health outcome (17 studies) followed by the studies of under-five mortality (including neonatal, post-neonatal, infant and child) (14 studies). Figure 2 presents a summary of the included studies by the number of publications, types of health outcomes and year of publication (Figure 2).

### 3.3. Quality and Risk of Bias Assessment Results

Based on the study-level RoB assessment, no study scored low RoB. Eighteen out of 50 (36%) analyses had moderate RoB, 16/50 (32%) analyses had serious RoB, and the remaining 16/50 (32%) analyses had critical RoB. The main contributors to the relatively high proportion of studies with serious and critical RoB at the study-level were inadequate control of known or potential confounders, methods of exposure assessment and outcome ascertainment, and misclassification of exposure. Based on the item-level RoB assessment, three of the seven domains, where all the studies included in this review scored low RoB, were on the selection of participants, missing data and selection of reported results, whilst confounding and the classification of exposure (i.e., separating households into those using relatively clean and relatively polluting fuels) were the two domains that resulted in serious or critical RoB. The results of RoB assessment and grading for each study and item-level are presented in the Appendix A.

### 3.4. Statistical Methods

As Table 1, Table 2, Table 3, Table 4, Table 5 and Table 6 highlight, there were inconsistencies in the ways these studies form predictor variable (cooking fuels)—some authors dichotomized it into polluting and clean; others categorised it into different multinomial forms. As a result, there were variations in the definition of exposed and non-exposed (comparator) households across studies. Though there were big dissimilarities across studies in the numbers and types of covariates adjusted, factors related to child, parental, household, health service and environment were observed. All studies of respiratory illnesses utilized the logistic regression approach (Table 1). Of 14 studies of all-cause under-five mortality, 9 used logistic regression, 4 used the Cox regression technique, and 1 used Poisson regression (Table 2). All studies of childhood nutritional status used logistic regression (1 binary and 4 multinomial) (Table 3). Linear regression was used by 3 studies of birthweight (Table 4).

### 3.5. Under-Five Respiratory Health Studies

Seventeen out of 50 (34%) analyses reported the association between cooking fuel use and ARI in children. Based on maternal recall, these studies ascertained ARI using a range of survey-based methods (Table 1). These ascertainment methods included (i) cough accompanied by short and rapid breathing which was coupled with a problem in the chest (7/17 studies), (ii) cough accompanied by short and rapid breathing (the associated problem in the chest not included) (9/17 studies), and (iii) difficulty in breathing and chest-related congestion and blocked nose and sought-after treatment (1/17 study). However, one study used two of these ascertainment methods (one as a robustness check) [47], all based on one method only. Twelve out of 17 (71%) studies reported an adverse association, while all the remaining studies found no effect. Eight out of 17 (47%) studies had moderate RoB, of which six reported positive associations. All the studies (three) that had serious RoB, and 3/6 studies that had critical RoB, reported positive associations. All the remaining studies reported no effect (Table 1).

### 3.6. Studies of All-Cause Under-Five Mortality

Fourteen out of 50 (28%) analyses reported associations between exposure to smoke from cooking fuels and all-cause child mortality (expressed as odds ratio, relative risk and hazard ratio) (Table 2). Under-five mortality in the DHS is collected based on a synthetic cohort life table approach, which is used to collect the probability of mortality for a small age segments and then combined into the standard age segments [67]. Most of these studies also presented the results of subgroup analysis, based on age, and for neonatal (0–28 days), post-neonatal (1–11 months), child (12–59 months) and under-five (0–59 months). Eleven (78%) studies reported positive associations between cooking fuel use and death in at least one of these subgroups, while the remaining three studies reported no association. The greatest odds of mortality were observed in neonates (age during the first 28 days of life) that were living in households where cooking was done by polluting fuel, which was reported by Naz and colleagues [38] (OR: 2.67, 95% CI: 1.47, 4.82), compared to households primarily using low-emission fuels. Additionally, one pooled analysis that collated data from 47 countries [60] found an adverse association between cooking with kerosene (RR: 1.34, 95% CI: 1.18, 1.52), and solid fuels (RR: 1.24, 95% CI: 1.14, 1.34) and neonatal death. Based on RoB assessment, 6/7 studies that had moderate RoB reported positive associations. Additionally, 3/4 studies that had serious RoB and 2/3 studies that had critical RoB reported positive associations. All the remaining studies reported no effect (Table 2).

### 3.7. Childhood Nutritional Status Studies

Five out of 50 (10%) analyses assessed the association between cooking fuel use and nutritional status (stunting and anaemia) in children (Table 3). These studies used the recorded age and height of the children to ascertain stunting, and haemoglobin count of the children to ascertain anaemia. These outcome ascertainment methods were robust compared to the methods used to ascertain the other health outcomes included in this review (Appendix A). All of these studies categorised kerosene under the low polluting (comparator) group. Three of these studies reported moderate to severe stunting and anaemia relative risk ratios among children living in households where cooking was done with only biomass (wood, crop waste, or dung) [33,65,66], while the remaining two studies reported no effect. Four out of five studies had serious RoB, whilst the remaining one had critical RoB. One of the studies that labelled under serious RoB and a study with critical RoB reported no association (Table 3).

### 3.8. Birthweight Studies

Five out of 50 (10%) analyses described the association of cooking fuel use and birthweight of children (Table 4). These studies used different forms of birthweight data, including weight retrieved from health card (recorded at birth), weight from maternal recall, and child’s size at birth (very large, large, average, below average and small), as judged by mothers at the time of interview. One of these studies used birthweight retrieved from the health card only [25], while the remaining studies used more than one of these birthweight data. All birthweight-related studies took households that were using electricity, LPG, or natural gas as a reference (i.e., unexposed to polluting fuels) group. Three out of five studies reported adverse associations, while the remaining two reported no effect. Among these studies, a study which was based on a relatively robust source of outcome information (weight retrieved only from health card) reported that exposure to three polluting fuels was associated with reduced birthweight [25], including kerosene (OR: 1.51, 95% CI: 1.08, 2.12), biomass (OR: 1.51, 95% CI: 1.08, 2.12), and coal (OR: 1.57, 95% CI: 1.03, 2.41). In contrast, another study that used the same source of information on the outcome (conducted separate analysis) [59] found no such association. Overall, four of five studies had serious RoB, and the remaining one had moderate RoB. Of the three studies that reported adverse association, two of them had serious RoB, and the other one had moderate RoB (Table 4).

### 3.9. Pregnancy and Birth Complications Related Studies

Four out of 50 (8%) analyses reported the association between cooking fuel use and pregnancy and birth complications. These studies included stillbirth (three studies) [23,24,32] and preeclampsia/eclampsia (one study) as outcomes [35]. These studies were based on self-report (maternal recall). Two of these studies (one of the stillbirth study and the preeclampsia/eclampsia study) [32,35] reported an adverse association, while the remaining two studies reported no association between solid fuel use and stillbirth. All of the stillbirth-related studies had critical RoB, and the remaining one had serious RoB (Table 5).

### 3.10. Other Health Outcomes (Five Studies)

Table 6 presents the details of studies on other outcomes in addition to those described in the preceding sections. We identified five studies (10%), and the outcomes included reduced body weight in adult women [53], elevated blood pressure in pre-menopausal women [64], self-reported asthma among adult men and women (aged 20–49 years) [36], and in persons aged ≥60 years [34] and tuberculosis in adults [30]. Two studies were rated as having moderate RoB, while the rest were rated as having critical RoB. All of the five studies reported positive associations (Table 6).

## 4. Discussion

The Demographic and Health Survey (DHS) is the largest and longest-running source of data on exposure to household air pollution (HAP) in low- and middle-income countries (LMICs), where the availability of vital registration statistics and other administrative records are often scarce. The DHS has a number of attractive features, including national coverage, high quality interviewer training, strong fieldwork coordination, standardised procedures across countries, and high response rates [11,67]. The use of uniform survey instruments allows for the comparison of health and health-related information both within and between countries. In most countries in which data are collected, surveys are repeated periodically, allowing methods and scope to be improved as new data needs and priorities emerge [11,68].

In environmental health research, the survey has seen strongest uptake in studies on water, sanitation and hygiene (WASH) [69] and HAP. Moreover, since 2000, DHS surveys have also collected spatial information on anonymised sampling locations and have been linked to environmental and sociodemographic datasets, both within data supplied by the DHS and by end-users [70,71]. Given the importance of location on exposure to environmental hazards, which can be spatially heterogeneous over small areas, this information has substantial potential to continue to enable a wider variety of studies, and different analytical methods. For studies of HAP from cooking, this can allow researchers to control for potential relevant spatial covariates in their analyses (e.g., air pollutants, temperature, humidity and rainfall, to name a few).

In this review, 45 studies that investigated the cross-sectional associations between cooking fuel use and 11 different health outcomes from 57 countries (there were studies that included more than one country) were included. In total, 36 (80%) studies reported statistically significant adverse associations between using polluting fuels and health outcomes.

### 4.1. Limitations of DHS Surveys (with Respect to HAP and Health)

The DHS surveys collect data mainly from self-reports through face-to-face interviews, which are prone to reporting and recall biases. Specific to the current review, for example, the DHS collects cooking fuel, which is widely used as a proxy to estimate HAP, although non-cooking sources (lighting and heating) are not elicited. The survey uses the 11-item classification of cooking fuel: electricity, liquid petroleum gas (LPG), natural gas, biogas, kerosene, coal/lignite, charcoal, wood, straw/shrubs/grass, crop waste, and animal dung. The question is: “what type of fuel does your household mainly use for cooking?” followed by the above list of fuels. This cooking fuel-related question has several limitations. First, it does not allow multiple responses that could help to capture fuel stacking, which is a common practice in LMICs and in rural settings where wood, dung, charcoal, kerosene, LPG and other fuels used in combination [72,73,74]. Second, despite the fact that households use polluting fuels, including coal, biomass and kerosene for heating and lighting [75,76], this information was not collected by the DHS surveys. Third, different stoves using the same fuel can vary markedly in their emission characteristics [1], but this level of detail is not collected in the DHS. Cooking emissions also affect outdoor air quality, which is beyond the DHS’ remit, but presents an important challenge for researchers attempting to select covariates in their analyses of cooking fuel and health associations. These factors may underestimate the level of exposure and the resultant disease burden in those regions. Moreover, many authors in the included studies mentioned that the survey they lacked some important variables, as measured by indices, such as completeness, detail and absence. For example, a study that pooled data from countries in sub-Saharan Africa indicated a large number of missing observations on cooking fuel (50.3%), on cooking location (53.4%) and on stove ventilation (59.1%) [63]. The DHS surveys are repeated at approximately five-yearly intervals, but the survey protocol and aims do not seek to target the same households for follow-up in each successive repeat, and longitudinal analyses are, therefore, not possible. Health researchers have performed numerous cross-sectional secondary analyses, but this is an important limitation of the survey in the context of epidemiological studies.

In terms of the health outcomes identified in this review, validation studies have shown that the tool used to ascertain ARI in children has limitations in discerning episodes of the real ARI from simple cough/cold accurately [77,78]. In addition, the studies we reviewed defined ARI in children differently. Since many births in the DHS regions are conducted at home, the weight at birth was missing for a substantial number of births in the dataset [63]. As a result, three out of five studies included in our review combined three sources of birthweight information in their analyses: birthweight retrieved from birth cards, birthweight recalled by mothers, and child’s size at birth (as reported by mothers using an ordinal scale). Whether this is more valid than using one source of information is unknown, and highlights the broader challenges faced by researchers and the often-ad hoc decisions that make direct comparison between studies difficult, even when using the same data. We reviewed 14 studies of child mortality (neonatal, post-neonatal, infant, child and under-five) and three studies on risk of stillbirth, which depend on accurate age and date of birth and death records but could be affected by age aggregation and inconsistencies in reporting [79,80]. Although anthropometric information in the DHS surveys is collected objectively with standardized methods, exposure misclassification (mixing relatively clean and relatively polluting fuels in assigning households to exposed and reference groups) was the main reason studies of childhood stunting and anaemia were rated as having serious RoB.

In general, inadequate control for potentially important covariates, such as ambient air pollutants (e.g., particulate matter, nitrogen oxides, ozone and carbon monoxide), was widely observed across studies. Most of these factors have been identified as important, either independently or as a confounder for respiratory health, mortality, nutritional status, and adverse pregnancy and birth outcomes. This gap might happen because of the fact that data on these ambient air pollutants do not exist in the existing DHS dataset, but can be linked from other sources using spatial information for sampling locations. Moreover, the covariates that are collected in many DHS surveys (e.g., environmental temperature and humidity) and known confounders for some of the child health studies (e.g., ARI and nutrition) [81,82] were not adequately considered in most studies. These variables are available in the majority of DHS surveys that were collected after 2000. Only two studies had attempted to control seasonal variations by taking the data collection/interview period as a season in a year in their analyses [45,63]. As a result, the majority of included studies were found to have moderate to critical RoB for the confounding domain in the item-level RoB assessment. Taking the confounding effect of these variables (at least those that are collected in the DHS surveys) into account could help to reduce such RoB. The misclassification of exposure status was also widely observed. More than one-third of the included studies classified kerosene under clean (low emission) fuels. However, the WHO, in its indoor air quality guidelines, regarded kerosene as a polluting fuel in 2014 [1]. One-third of those studies that considered kerosene as a low emission fuel were published after these guidelines. In terms of outcome ascertainment methods, with the exception of height and weight (used to determine nutritional status) and haemoglobin count (used to determine anaemia), all measures in the DHS are captured from maternal/caregivers recall (e.g., ARI and mortality in children) and self-reporting (e.g., pregnancy and birth complications). As a result, bias related to the misclassification of the outcome could be significant. In addition, studies included in this review used different definitions for the same outcome (e.g., ARI in children and weight at birth). This risk of bias could be minimised by using any available guideline; for example, the revised version of pneumonia guidelines for ARI in children [83].

### 4.2. Strengths and Limitations of this Review

We reviewed studies based on the DHS because it is an important, often sole, source of data on HAP and health outcomes in LMICs. We used a new risk of bias assessment tool, which is appropriate for environmental exposure studies, and assessed risk of bias both at study level and at outcome level [22]. In addition to the standard bibliographic databases, we searched evidence under the DHS publication portal. About 90% of the studies in our review were new (not included in previous reviews) and our review entirely focused on epidemiological studies obtained from further analyses of DHS data. We hope our findings and suggestions are helpful to end users of DHS data, as well as the large global team tasked with designing and administering the crucial work of the DHS program. To that end, we have summarised our impression of the main challenges, together with some suggestions for how they might be addressed, in Table 7.

This review has several limitations. One of the limitations of our review is that we have included peer-reviewed research articles published only in the English language. As the DHS surveys are also conducted in non-English speaking countries and the survey information is widely used to develop government documents (e.g., working papers), we might have missed grey literature and some records published in languages other than English. However, we do not think this would alter our substantive findings, as the data collected under the DHS program, and used by researchers, are similar. Finally, for practical reasons, the RoB assessment was undertaken by one of us, and despite using a standard process for assessing each article’s RoB, errors or omissions may mean the RoB has been under- or over-estimated for individual studies.

## 5. Conclusions

Although the HAP exposure assessment is not the main focus of the DHS, it is the main, often only, source of information in many low- and middle-income countries. It remains an important resource, and researchers have used it to perform important cross-sectional analyses on HAP from cooking fuels and health in 57 countries. By restricting our focus to DHS-based studies, our review allowed for greater granularity in our assessment of DHS-specific issues. Through that, we found that an appreciable proportion of the studies we reviewed have potential for serious or critical RoB, due mainly to the inherent limitations of what information can feasibly be collected in such a large-scale survey, and methodological choices by end-users. This relates to confounder control, exposure misclassification, and outcome ascertainment. We underscore to researchers’ need to bear these issues in mind in their analyses. Moreover, we also offer some pragmatic suggestions through which DHS data could be even more amenable to studies of household fuel use and health, and reduce the RoB, without being onerous to collect or analyse. We hope our review is useful for researchers and others working with DHS data.

## Figures and Tables

**Figure 1 ijerph-18-01411-f001:**
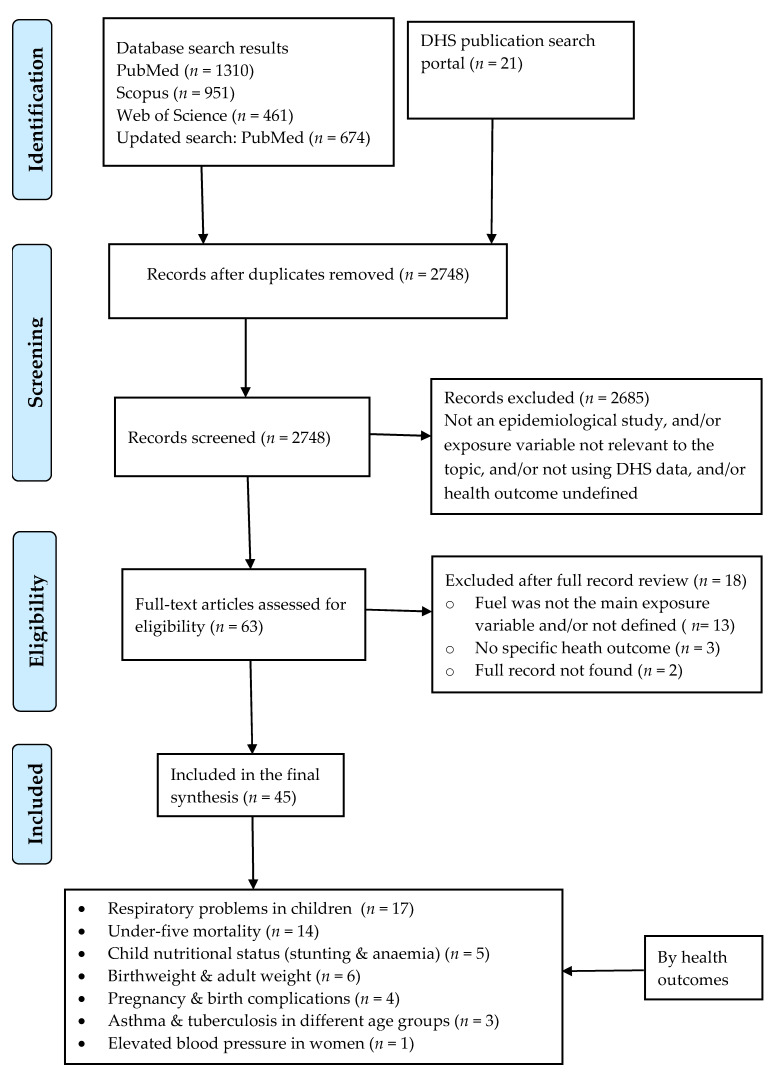
The flow of studies from identification to data extraction from databases based on the PRISMA guidelines. Note: three articles reported seven health outcomes, together.

**Figure 2 ijerph-18-01411-f002:**
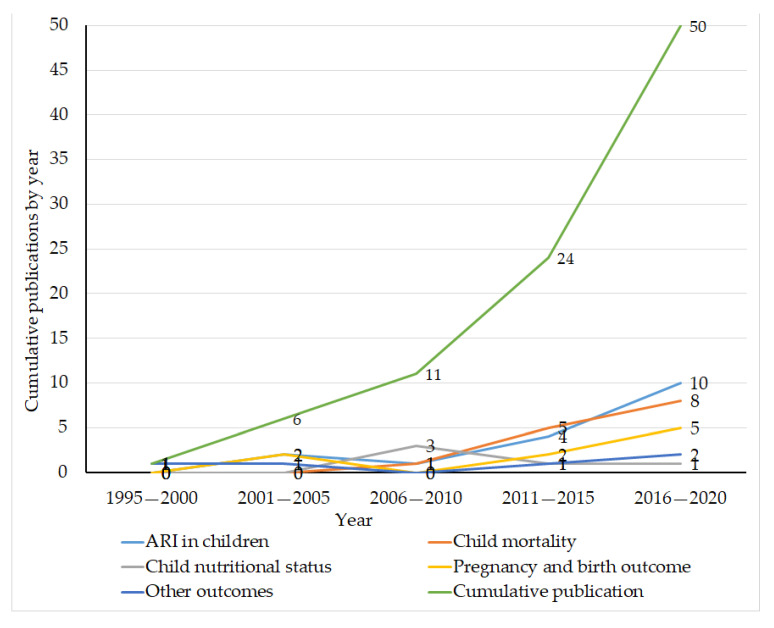
Number of publications reporting effects of household air pollution on health from DHS data, by health outcomes and year of publication, 2020. Note: pregnancy and birth outcomes: birthweight, stillbirth, and preeclampsia/eclampsia. Other health outcomes: asthma (2 studies), maternal body weight (1 study), tuberculosis (1 study) and premenopausal blood pressure (1 study). ARI: acute respiratory infection.

**Table 1 ijerph-18-01411-t001:** DHS based evidence on the association between cooking fuel use and acute respiratory infection in children.

Authors and Publication Year	Country and Survey Year	Sample Size	Exposed Group	Comparator Group	Outcome: Case Definition	Covariate Adjusted (See Footnote)	Statistical Method and Main Results	Risk of Bias Score
Mondal et al. (2020)	India 2015–16	247,743	Biomass fuels: kerosene, coal/lignite, charcoal, wood, crop waste, straw/shrubs/grass, animal dung and others	Clean fuels: electricity, LPG, natural gas, and biogas	ARI in children aged <5 year: cough accompanied by short, rapid, or difficult breathing that is chest related	1, 2, 3, 14, 15, 30, 31, 45, 47, 49, 52, and 78a	Logistic regression OR, 95% CI: 1.10 (1.01–1.20)	Moderate
Naz et al. (2020)	Pakistan 3 waves (2006–2018)	8307 for 2006 10,805 for 2012 9807 for 2017	Polluting fuel: kerosene, wood, straw, shrubs, grass, animal dung, coal, charcoal, crop waste, and others	Clean fuel: LPG, electricity, biogas, and natural gas	Pneumonia in children aged <5 year: difficulty in breathing and chest-related congestion and blocked nose and sought-after treatment	1, 2, 4, 5, 14, 15, 17, 20 and 56	Logistic regression OR, 95% CI: PDHS 2006–07: 1.26 (0.98, 1.61) PDHS 2012–13: 1.16 (1.00, 1.34) PDHS 2017–18: 1.30 (0.99, 1.39)	Moderate
Woolley et al. (2020)	Uganda 2016	13,266	Households using wood for cooking	Households using charcoal for cooking	ARI in children aged <5 year: cough and short rapid breaths/difficulty Severe ARI in children aged <5 year: cough, short rapid breaths/difficulty breathing and fever	1, 1a, 1b, 2, 3, 6, 14, 15, 19, 47, 49, 50, 52, 53 and 67	Logistic regression OR, 95% CI: ARI: 1.36 (1.11–1.66) and Severe ARI: 1.41 (1.09–1.85), in wood users compared to charcoal users.	Moderate
Budhathoki et al. (2020)	Nepal 3 waves (2006–2016)	5139 for 2011 4887 for 2016	Polluting fuels: kerosene, wood, straw, shrubs, grass, animal dung, coal and charcoal	Clean fuels: electricity, LPG, biogas and natural gas	Pneumonia in children aged <5 year: cough accompanied by (1) short, rapid breathing that is chest-related and/or (2) difficult breathing that is chest-related (definition from NDHS report).	4, 7, 14, 15, 24, 47, 49, 56	Logistic regression (authors reported their results as A relative risk (RR) RR, 95% CI: NDHS 2011: 1.19 (0.72, 1.98) NDHS 2016: 1.98 (1.01, 3.92) Authors declared that data not available for the 2006 NDHS on fuel use	Moderate
Rana et al. (2019)	Afghanistan 2015	27,565	Solid fuel: coal, lignite, charcoal, wood, animal dung, straw/shrubs/grass, and kerosene	non-solid fuel: electricity, LPG, natural gas, and biogas	ARI in children aged <5 year: cough with shortness of breathing or difficulty in breathing	1, 2, 6, 14, 15, 16, 23, 51 and 52	Mixed-effect Poisson regression Prevalence Ratio, 95% CI: 1.10 (0.98, 1.23)	Moderate
Khan et al., 2018	Pakistan, 2012–13 PDHS	11,040	Polluting fuels: wood, dung, charcoal, coal, shrubs/grass/straw, or kerosene	Cleaner fuel: natural gas, LPG, biogas, or electricity	ARI symptoms in children aged <5 year: (1) Cough accompanied by short and rapid breathing, and (2) Cough, accompanied with short and rapid breathing, coupled with a problem in the chest	1, 2, 3, 5, 6, 14, 15, 16, 45 and 47	Logistic regression OR, 95% CI: cough with short and/or rapid breathing: 1.51 (1.03, 2.21) cough with short and/or rapid breathing and problem in chest: 1.37 (0.84–2.24)	Moderate
Capuno et al. (2018)	Philippines, 2013 NDHS	5442	Kerosene or solid fuels (coal, lignite, charcoal, wood, crop waste, dung or shrubs/grass/straw)	Electricity, LPG, natural gas or biogas	ARI Symptoms in children aged <5 year: Cough accompanied by short, rapid breathing or difficulty in breathing as a result of a problem in the chest	2, 3, 4, 12, 13, 14, 15, 30, 31, 32, 34, 35, 37, 50, 52, 53 and 60	Logistic regression with propensity score marching: Average treatment effect on the treated (ATT) NN1: −0.024, *p* < 0.1 NN3: −0.021, *p* < 0.1 NN5: −0.022, *p* < 0.1	Critical
Akinyemi et al. (2018)	Nigeria, 2003–2013 NDHS	5445 (2003) 24,975 (2008) 28,950(2013)	Unclean fuel: coal, lignite, charcoal, wood, kerosene, dung or shrubs/grass/straw	Clean fuel: electricity, LPG, natural gas or biogas	ARI symptoms in children aged <5 year: Cough in the last 2 weeks and if a cough was accompanied by short rapid breaths	1, 3, 7, 14, 15, 24, 47, 49, 50, 51, 52, 54, 56, 57 and 67	Logistic regression OR, 95% CI: 0.99 (0.98, 1.00) for 2003, 1.00 (0.99, 1.01) for 2008, and 1.44 (0.66, 3.13) for 2013	Moderate
Khan et al. (2017)	Bangladesh, 2007–2014 BDHS	22,789	Solid fuel: coal, lignite, charcoal, wood, dung, straw/shrubs/grass, crop waste, and others	Clean fuel: electricity, LPG, natural gas, or biogas	ARI in children aged <5 year: Infection in the nose, trachea or lungs that interfere normal breathing	2, 6, 14, 15, 50, 51 and 52	Logistic regression OR, 95% CI: Inside vs. outside cooking 1.18 (1.08, 1.33) Clean vs. solid fuel 1.07 (0.95, 1.20)	Critical
Daniel (2016)	Cameron and Gabon, 2011 CDHS and 2012 GDHS	Cameroon 5 821 Gabon 1952	1. Only biomass fuel: wood, crop waste, dung, straw/shrubs/grass, and 2. Other fuels: kerosene, coal (lignite) or charcoal	Electricity, gas and/or biomass fuel	ARI in children aged <5 year: Cough, accompanied with short and rapid breathing, coupled with a problem in the chest	1, 7, 57, 47, 50, 14, 15, 45, 3, 61 and 52	Logistic regression OR, 95% CI: Cameroon: Only biomass: 5.62 (1.29, 24.44) Other fuel: 4.13 (0.51, 33.57) Gabon: Only biomass: 1.71 (0.98, 2.97) Other fuel: 3.99 (1.46, 10.91)	Critical
Wichmann et al. (2015)	South Africa, 1998 SADHS	4679	Polluting fuel: wood, dung, paraffin, charcoal, or combination of these with clean fuel	Clean fuel: electricity, LPG or natural gas, exclusively	ALRI in children aged <5 year: Cough, accompanied by short and rapid breathing	1, 2, 3, 14, 15, 31 and 53	OR, 95% CI: 1.27 (1.05, 1.55)	Serious
Buchner et al. (2015)	SSA countries, DHS conducted from 2000 to 2011	56,437	1. Kerosene 2. Coal and charcoal 3. Wood 4. Lower-grade biomass fuels (shrubs, crop waste or dung)	Clean fuels: electricity, gas, biogas plus responses of no food cooked in house	ALRI in children aged <5 year: Cough and short rapid breath or problems in the chest or a blocked or running nose	1, 3, 6, 5, 7, 16, 15, 34, 48, 55, 47, 45, 57, 52, 60, 67, 68, 30, 37 and 38	Mixed model logistic regression OR, 95% CI: Kerosene: 1.23 (0.77, 1.95) Coal/Charcoal: 1.35 (1.06, 1.72) Wood: 1.32 (1.04, 1.66) Lower-grade biomass: 1.07 (0.69, 1.66)	Moderate
Acharya et al. (2015)	Nepal, 2011 NDHS	4773	Solid fuel: wood, animal dung, straw, shrubs, grass, crop waste, coal (lignite) or charcoal	Cleaner fuel: LPG, biogas, electricity, natural gas or kerosene	ARI in children aged <5 year: Cough accompanied by short/rapid breath and problem in chest	1, 2, 3, 15, 16, 50, 51, 52, 53 and 65	Logistic regression OR, 95% CI: 1.79 (1.02, 3.14).	Serious
Patel et al. (2013)	India, INFHS conducted from 1992 to 2006	36,254	1. Highly polluting fuel: wood, crop waste, dung, or straw 2. Medium polluting fuel: coal/lignite, charcoal, or kerosene	Low polluting fuel: LPG, natural gas or electricity	ALRI in children aged 0–35 months: Cough accompanied by rapid breathing	14, 15, 24, 1, 2, 4, 52, 31, 50, 51 and 25	Logistic regression OR, 95% CI: Highly polluting fuels: 1.48 (1.08, 2.03) for 1992 1.54 (1.33, 1.77) for 1999 1.53 (1.21, 1.93) for 2006 Medium polluting fuels: 1.39 (1.01, 1.92) for 1992 1.47 (1.22, 1.76) for 1999 1.31 (0.92, 1.88) for 2006	Critical
Kilabuko et al. (2007)	Tanzania, 2004–2005 TDHS	5224	Biomass fuels: firewood, straw, dung or crop waste	Charcoal or kerosene	ARI in children aged <5 year: Cough together with short and rapid breathing	14, 15, 1, 2, 50, 51 and 52	Logistic regression OR, 95% CI: 1.01 (0.78, 1.42)	Critical
Mishra et al. (2005)	India, 1998–1999 NFHS	29,768	1. Biomass fuels: wood, crop waste, or dung 2. Mixed fuels: biomass fuels, charcoal and coal	Cleaner fuels: electricity, LPG, biogas or kerosene	ARI in children aged 0–36 months: Cough, with breath faster than usual with short, rapid breaths	1, 2, 3, 14, 15, 30, 31, 7, 49, 57, 47, 45, 52, 50, 51 and 53	Logistic regression OR, 95% CI: Biomass fuel: 1.58 (1.28, 1.95) Mixed fuel: 1.41 (1.17, 1.70)	Serious
Mishra (2003)	Zimbabwe, 1999 ZDHS	3559	1. Highly polluting fuels: wood, dung, or straw 2. Medium polluting fuels: kerosene or charcoal	Low polluting fuels: LPG, natural gas, or electricity	ARI in children aged <5 year: Cough accompanied by short and rapid breathing	1, 2, 3, 7, 14, 15, 30, 51 and 52	Logistic regression OR, 95% CI: High polluted: 2.20 (1.16, 4.19) Medium polluted: 1.33 (0.64, 2.77)	Critical

All effect sizes are from adjusted result. LPG—liquefied petroleum gas, U5—under-five, OR—odds ratio, RR—relative risk, ARI—acute respiratory infection, ALRI—acute lower respiratory infection, DHS—Demographic and Health Survey (the first one or two letters represent country name initials), NFHS—National Family Health Survey. DHS surveys are cross-sectional by design and ARI is asked for the period "anytime during the last 2 weeks". Covariate adjusted: **child related factors**: 1. Child’s age, 2. child’s gender, 3. Child’s birth order, 4. Birth weight/birth size, 5. Child’s vaccination status, 6. Child’s breastfeeding status, 7. Child’s nutritional status, 8. Parity, 9. Pregnancy type (single/multiple), 10. Inter-birth interval, 11. Year of birth, 12. Child live with mother, 13. Own child or grandchild, 1a. Vitamin A supplementation, 1b. Mode of delivery. **Individual/family related factors**: 14. Mother’s age, 15. Mother’s education status, 16. Maternal smoking status, 17. Maternal alcohol consumption status, 18. Mother’s body mass index, 19. Took iron during, 20. Mother’s anemia status pregnancy, 21. Took malaria drug during pregnancy, 22. Pregnancy termination history, 23. Mother’s occupation, 24. Maternal working status, 25. Media exposure, 26. Mother controlled by husband, 27. Mother physically abused by husband, 28. Mother humiliated by husband, 29. Mother’s perception of medical care, 30. Religion, 31. Ethnicity, 32. Maternal marital status, 33. Mather’s occupation, 34. Father’s education, 35. Father’s age, 36. Father’s smoking status, 37. Gender of household head, 38. Age of household head, 39. Smoking status, 40. Age, 41. Gender, 42. Marital status, 43. Educational status. **Household factors**: 44. Number of under-five children, 45. Crowding, 46. Number of sleeping room, 47. Cooking location, 48. Stove ventilation, 49. Presence of smoker in the household, 50. Place of residence, 51. Region of residence, 52. Household wealth index, 53. Family size, 54. Drinking water source, 55. Time to water source, 56. Latrine status, 57. Housing material, 58. Presence of window, 59. Access to electricity, 60. Having health insurance, 61. Utilization of health care service, 62. Food security. **Health service and environmental related factors**: 63. Number of ANC attended, 64. Place of delivery, 65. Ecological zone, 66. Survey year, 67. Season of interview, 68. Geographic location, 69. Country, 70. Acute respiratory infection status of a child, 71. Diarrhea status of a child, 72. Fever status of a child, 73. Stunting status of a child, 74. Wasting status of a child, 75. Malaria status of a child, 76. Respondent’s diabetes status, 77. Respondent’s asthma status, 78. Month of interview, 78a. History of TB contact.

**Table 2 ijerph-18-01411-t002:** DHS data-based evidence on the association between cooking fuel use and under-five mortality (all-cause).

Authors and Publication Year	Country and Survey Year	Sample Size	Exposed Group	Comparator Group	Case Definition	Covariate Adjusted (Table 1 Footnote)	Statistical Method and Main Results	Risk of BiasScore
Samuel et al. (2018)	Nigeria, 2013 NDHS	10,983	Solid fuel: wood, charcoal and dung, in a kitchen inside the house	Non-solid fuel: electric, gas and kerosene, in a kitchen inside the house	Under-five mortality during the last five years preceding the survey	15, 52, 50 and 51	Logistic regression OR, 95% CI: 1.23 (0.98, 1.54)	Critical
Nisha et al. (2018)	Bangladesh, 2004, 2007, 2011, and 2014 BDHS	35,052	Polluting fuels: kerosene, coal/lignite, charcoal, wood, crop waste, dung or straw/shrubs/grass	Clean fuels: electricity, LPG, natural gas, and biogas	Death between the ages of 0 and 6 days	3, 14, 15, 18, 24, 50, 52, 47 and 66	Logistic regression OR 95% CI: Early neonatal: 1.46 (1.01, 2.10)	Critical
Naz et al. (2018)	Nepal, 2001–2011 NDHS	17,780	Polluting fuels: kerosene, coal/lignite, charcoal, wood, straw/shrubs/grass, crop waste or dung	Clean fuels: electricity, LPG, natural gas and biogas	Under-five mortality within 5 years prior to the survey	2, 6, 14, 15, 24, 50, 52, 47, 65, 57 and 8	Logistic regression OR, 95% CI: Neonatal: 2.67 (1.47, 4.82) Post-neonatal: 1.61 (0.67, 3.87) Child: 1.29 (0.33, 4.99) Under-five: 2.19 (1.37, 3.51)	Moderate
Owili et al. (2017)	23 SSA countries, DHS conducted from 2010 to 2014	783,691	1. Charcoal 2. Other biomass: wood, straw/shrubs/grass, crop waste, or dung 3. Other polluting fuel: coal/lignite or kerosene	Clean fuel: electricity, natural gas, biogas or LPG	All-cause mortality of under-five children within 5 years prior to the survey	47, 69, 50, 2, 6, 14, 15, 53, 44, 52, 49, 33 and 23	Cox regression HR, 95% CI: Charcoal: 1.21 (1.10, 1.34) Other biomass: 1.20 (1.08, 1.32) Other fuel: 1.01 (0.90, 1.14)	Moderate
Naz et al. (2017)	Pakistan, 2013 PDHS	11,507	Polluting fuels: kerosene, coal/lignite, charcoal, wood, straw/shrubs/grass or dung	Clean fuels: electricity, LPG, natural gas and biogas	Neonatal, post-neonatal, child and under-five mortality	1, 2, 6, 14, 15, 24, 16, 50, 52, 57 and 47	Logistic regression OR, 95% CI: Logistic regression Neonatal: 1.09 (0.77, 1.54) Post-neonatal: 1.31 (0.75, 2.27) Child: 1.98 (0.75, 5.25) Under-five: 1.22 (0.92, 1.64)	Moderate
Khan et al. (2017)	Bangladesh, 2007–2014 BDHS	22,789	Solid fuels: coal, lignite, charcoal, wood, straw/shrubs/grass, crop waste, dung and others	Clean fuel: electricity, LPG, natural gas, and biogas	Neonatal, infant and under-five mortality	2, 6, 14, 15, 50, 51 and 52	Logistic regression OR, 95% CI: Neonatal: 1.23 (0.97, 1.55) Infant: 1.15 (0.94, 1.42) Under-five: 1.11 (0.91, 1.35)	Serious
Naz et al. (2016)	India, 1992–2006 NFHS	166,382	polluting fuels: kerosene, coal/lignite, charcoal, wood, straw/shrubs/grass, crop waste or dung	Clean fuels: electricity, LPG, natural gas and biogas	Neonatal, post-neonatal, child and under-five mortality	1, 6, 14, 15, 24, 16, 50, 52, 57, 66 and 47	Logistic regression OR, 95% CI: Neonatal: 1.23 (1.09, 1.39) Post-neonatal: 1.42 (1.19, 1.71) Child: 1.42 (1.05, 1.91) Under-five: 1.30 (1.18,1.43)	Moderate
Akinyemi et al. (2016)	15 SSA countries, DHS conducted from 2010 to 2014	143,602	Solid fuel: coal, lignite, charcoal, wood, straw/shrubs/grass, crop waste or dung	Non-solid fuel: electricity, LPG, natural gas, biogas and kerosene	Infant and child mortality	2, 3, 4, 10, 14, 15, 23, 16, 25, 49, 50, 52 and 53	Cox regression HR, 95% CI: Smoking + solid fuel: 1.59 (1.26, 1.99) Smoking + non-solid fuel: 0.86 (0.44, 1.68) No smoking + solid fuel: 1.44 (1.18, 1.76)	Serious
Naz et al. (2015)	Bangladesh, 2004, 2007 and 2011 BDHS	18,308	Polluting fuels: kerosene, coal/lignite, charcoal, wood, straw/shrubs/grass, crop waste or dung	Clean fuels: electricity, LPG, natural gas, and biogas	Neonatal, infant and under–five mortality	6, 14, 15, 24, 52, 50, 57 and 66	Logistic regression OR, 95% CI: Neonatal: 1.49 (1.01, 2.22) Infant: 1.27 (0.91, 1.77) Under-five: 1.14 (0.83, 1.55)	Moderate
Kleimola et al. (2015)	47 Countries, DHS conducted from 2001 to 2012	774,638 neonates and 751,571 children	Kerosene and Solid fuels: coal, charcoal, and biomass such as wood, crop waste or dung	Clean fuels: electricity, LPG, natural gas, and biogas	Neonatal and child mortality	2, 3, 14, 15, 16, 52, 50 and 69	Poisson regression RR, 95% CI: Kerosene- neonatal: 1.34 (1.18, 1.52) solid fuel- neonatal: 1.24 (1.14, 1.34) Kerosene- child: 1.12 (0.99, 1.27) Solid fuel- child: 1.21 (1.12, 1.30)	Critical
Ezeh et al. (2014)	Nigeria, 2013 NDHS	30,726	Solid fuel: coal/lignite, charcoal, wood, straw/shrubs/grass, crop waste and dung	Non-solid fuels: electricity, LPG, natural gas, biogas, or kerosene	Neonatal, post-neonatal and child mortality	2, 6, 4, 14, 15, 24, 50 and 47	Cox regression HR, 95% CI: Neonatal: 1.01 (0.73, 1.26) Post-neonatal: 1.92 (1.42, 2.58) Child: 1.63 (1.09, 2.42)	Serious
Pandey et al. (2013)	India, 2005–6 NFHS	25,839	Solid fuel: coal/lignite, charcoal, wood, straw, shrubs, grass, crop waste, or dung	Other fuels: electricity, LPG, natural gas, biogas or kerosene	Women who had experienced death of at least one under-five child death	16, 14, 15, 30, 26, 27, 28, 35, 34, 52, 50, 48 and 58	Logistic regression OR, 95% CI: 1.23 (1.06,1.43)	Serious
Epstein et al. (2013)	India, 2005–6 NFHS	14,850	High-pollution fuels: (a) kerosene, (b) coal/coal lignite, (c) biomass fuels (wood, charcoal, crop waste, dung or straw/shrubs/grass)	Low-pollution fuels: LPG, natural gas or biogas	Death in the first 0–28 days of life	2, 4, 14, 15, 30, 16, 23, 18, 50, 51, 59, 64, 57, 54, 29, 63, 8 and 10	Logistic regression OR, 95% CI: Kerosene: 2.88 (1.18, 7.02) Biomass: 0.84 (0.39, 1.81) Coal: 24.15 (7.98, 73.12)	Moderate
Wichmann et al. (2006)	South Africa, 1998 SADHS	3556	Polluting fuels: wood, dung, coal or paraffin	Clean fuels: LPG, natural gas or electricity, exclusively	Mortality during 1–59 months of age	1, 2, 3, 6, 7, 10, 14, 54, 56, 45 and 11	Cox regression HR, 95% CI: 1.99 (1.04, 3.68)	Moderate

All effect sizes are from adjusted result. Both exposure and outcome ascertainment were through self-report. LPG—liquefied petroleum gas, OR—odds ratio, RR—relative risk, HR—hazard ratio, DHS—Demographic and Health Survey (the first one or two letters represent country name initials), NFHS—National Family Health Survey.

**Table 3 ijerph-18-01411-t003:** DHS data-based evidence on the association between cooking fuel use and under-five nutritional status.

Authors and Publication Year	Country and Survey Year	Unit of Analyses	Sample Size	Exposed Group	Comparator Group	Outcome Definition	Covariate Adjusted (Table 1 Footnote)	Statistical Method and Main Results	Risk of Bias Score
Dadras et al. (2017)	Nepal, 2011 NDHS	Under-five children	2262	High polluting fuels: wood, dung, straw or crop waste	Low polluting fuels: LPG, natural gas, biogas, electricity, kerosene, coal or charcoal	Stunting height-for age Z-score <−2 SD	1, 3, 4, 15, 16, 18, 47, 51, 52, 54, 31 and 62	Logistic regression OR, 95% CI: 1.13 (0.72, 1.76)	Critical
Machisa et al. (2013)	Swaziland, 2006–7 SDHS	Children aged 6–36 months	1150	Biomass fuel: coal, charcoal or wood with or without cleaner fuels	Cleaner fuels: LPG, natural gas, electricity and/or paraffin, exclusively	Anaemia and stunting in under 5 children	1, 2, 3, 10, 4, 14, 15, 18, 19, 20, 50, 51, 47, 52, 70, 71 and 72	Logistic regression (multinomial) RRR, 95% CI: Mild stunting: 1.1 (0.6, 2.0) Sever stunting: 1.4 (0.7, 2.7) Only descriptive result reported for anaemia	Serious
Kyu et al. (2010)	DHS conducted from 2003 to 2007 in 29 countries	Children aged 0–59 months	117,454	Households using wood, straw, dung or crop waste	Households using Electricity, natural gas, biogas or kerosene	Anaemia in children	1, 2, 14, 15, 16, 19, 20, 52, 69, 71, 72, 44, 73 and 74	Logistic regression (multinomial) RRR, 95% CI: Mild: 1.07 (1.01, 1.13) Moderate/sever: 0.99 (0.94, 1.05)	Serious
Kyu et al. (2009)	DHS conducted from 2005 to 2007 in 7 countries	Under-five children	28,439	Coal and biomass fuels such as wood, straw or dung)	Cleaner fuels: electricity, natural gas, biogas and kerosene	Stunting	1, 2, 4, 6, 14, 15, 16, 36, 44, 52 and 69	Logistic regression (multinomial) RRR, 95% CI: Stunting: 1.25 (1.08, 1.44) Severe stunting: 1.27 (1.02, 1.59)	Serious
Mishra et al. (2007)	India, 1998–99 NFHS	Children aged 0–35 months	29,768	1. only biofuels (wood, crop waste, or dung) 2. mix of biomass fuels and cleaner fuels (coal/lignite or charcoal)	Cleaner fuels: electricity, liquid petroleum gas, biogas, or kerosene	Stunting and anaemia	2, 3, 14, 15, 18, 20, 19, 30, 31, 70, 71, 75, 57, 51, 52, 45 and 49	Logistic regression (multinomial) RRR, 95% CI: Moderate-to-severe anaemia Only biomass fuel: 1.58 (1.28, 1.94) Mixed fuel: 1.36 (1.13, 1.63) Severe stunting Only biomass fuel: 1.90 (1.49, 2.42) Mixed fuel: 1.26 (1.00, 1.58)	Serious

All effect sizes are from adjusted result. Exposure status measured through self-report. LPG—liquefied petroleum gas, OR—odds ratio, DHS—Demographic and Health Survey (the first one or two letters represent country name initials), NFHS—National Family Health Survey.

**Table 4 ijerph-18-01411-t004:** DHS data-based evidence on the association between cooking fuel use and birthweight.

Authors and Publication Year	Country and Survey Year	Unit of Analyses	Sample Size	Exposed Group	Comparator Group	Outcome and Definition	Covariate Adjusted (Table 1 Footnote)	Statistical Method and Main Results	Risk of BiasScore
Milanzi et al. (2017)	Malawi, 2010 MDHS	Under-five children	9124	Highly polluting fuels: charcoal, wood, crop waste, straw or dung	Low pollution fuels: electricity, LPG or biogas	Birthweight from health card and size at birth recalled	2, 3, 14, 15, 18, 30, 52 and 50	Linear plus logistic regression OR, 95% CI: birthweight (continuous): 92 g (−320.4; 136.4) Size at birth (binary): 1.29 (0.34; 4.80)	Serious
Khan et al. (2017)	Bangladesh, 2007–2014 BDHS	Under-five children	22,789	Solid fuel: coal, lignite, charcoal, wood, straw/shrubs/grass, crop waste, dung or others	Clean fuels: electricity, LPG, natural gas or biogas	Birthweight from health card, maternal recall and size at birth	2, 6, 14, 15, 50, 51 and 52	Logistic regression OR, 95% CI: 1.33 (1.14–1.56)	Serious
Epstein et al. (2013)	India, 2005–6 NFHS	Under-five children (singleton recent birth)	14,850	Highly polluting fuels: (a) kerosene, (b) coal/lignite, (c) biomass fuels (wood, charcoal, crop waste, dung or straw/shrubs/grass)	Low polluting fuels: LPG, natural gas or biogas	Birthweight from health card	2, 14, 15, 16, 23, 18, 50, 51, 59, 54, 63, 8, 10 and 29	Logistic regression OR (95% CI): Kerosene: 1.51 (1.08, 2.12) Biomass: 1.51 (1.08, 2.12) Coal: 1.57 (1.03, 2.41)	Moderate
Sreeramareddy et al. (2011)	India, 2005–6 NFHS	Under-five children (most recent singleton births)	47,139	Highly polluting fuels: wood, straw, dung, and crop waste, kerosene, coal or charcoal	Low polluting fuels: electricity, LPG, natural gas or biogas	Birthweight from health card, from maternal recall and size at birth	2, 3, 20, 18, 14, 15, 16, 30, 52 and 50	Logistic regression Mean birthweight difference 73 g (2883.8 g vs. 2810.7g), *p* < 0.001) OR (95% CI): 1.07 (0.94, 1.22)	Serious
Mishra et al. (2004)	Zimbabwe, 1999 ZDHS	Under-five children (singleton births)	3331	Highly polluting fuels: wood, dung, or straw, and Medium polluting fuels: kerosene or charcoal (presented as a descriptive only)	Low polluting fuels: LPG, natural gas or electricity	Birthweight from health card or mother’s recall	2, 3, 14, 15, 30, 18, 19, 21, 51 and 52	Logistic regression OR (95% CI): from a health card −120 g (−301, 61) from recall −183 g (−376, 10)	Serious

All effect sizes are from adjusted result. Exposure status measured through self-report. LPG—liquefied petroleum gas, OR—odds ratio, DHS—Demographic and Health Survey (the first one or two letters represent country name initials), NFHS—National Family Health Survey.

**Table 5 ijerph-18-01411-t005:** DHS data-based evidence on the association between cooking fuel use and adverse pregnancy and birth outcomes.

**Authors and Publication Year**	**Country and Survey Year**		**Unit of Analysis**	**Sample Size**	**Exposed Group**	**Comparator Group**	**Outcome Definition**	**Covariate Adjusted (Table 1 Footnote)**	**Statistical Method and** **Main Results**	**Risk of Bias** **Score**
Nisha et al. (2018)	Bangladesh (2004–2014) BDHS		Singleton pregnancy	27,237	Polluting fuels: kerosene, coal/lignite, charcoal, wood, straw/shrubs/grass, crop waste, or dung	Clean fuels: electricity, LPG, natural gas, and biogas	Stillbirth: Foetal death in pregnancy of at least 7 or more months	3, 14, 15, 18, 24, 50, 52, 47 and 66	Logistic regression OR 95% CI: 1.25 (0.85, 1.84)	Critical
Khan et al. (2017)	Bangladesh (2007–2014) BDHS		Ever married women (10–49 year.)	22,789	Solid fuel: coal/lignite, charcoal, wood, straw/shrubs/grass, crop waste, dung or others	Clean fuel: electricity, LPG, natural gas or biogas	Stillbirth: Foetal death lasting seven or more months	2, 6, 14, 15, 50, 51 and 52	Logistic regression OR 95% CI: (OR 0.96: 0.73, 1.27)	Critical
Mishra et al. (2005)	India, 1998–99 NFHS		Ever married women aged 40–49 years	19,189	1. highly polluting group: only biomass fuels (wood, dung, or crop waste 2. medium polluting groups: mix of biomass fuels and cleaner fuels (coal/lignite or charcoal)	Only cleaner fuels: electricity, LPG, biogas, or kerosene	Stillbirth: delivery of a dead baby after the 28th week of pregnancy	16, 20, 15, 30, 57, 47, 31, 45, 52, 51, and 44	Logistic regression OR (95% CI): Highly polluting group: 1.44 (1.04, 1.97) risk of repeated incidence of stillbirth RRR (95% CI): Highly polluting group: 2.01 (1.11, 3.62)	Critical
Agrawal (2015)	India, 2005–2006 NFHS		Women aged 15–49 (who had live birth in the previous 5 years)	39,657	High and medium exposure group: biomass fuels such as, wood, crop waste, dung, straw/shrubs/grass, or solid fuels such as coal/lignite and charcoal	Low-exposure group: only cleaner fuels (kerosene, LPG/natural gas, biogas, or electricity)	Preeclampsia/Eclampsia: difficulty with vision during daylight, and swelling of the legs, body, or face and convulsions (not from fever)	44, 9, 22, 18, 16, 17, 76, 77, 20, 14, 15, 30, 52, 50 and 51	Logistic regression OR (95% CI): 2.21 (1.26, 3.87)	Serious

All effect sizes are from adjusted result. Exposure status measured through self-report. LPG—liquefied petroleum gas, OR—odds ratio, DHS—Demographic and Health Survey (the first one or two letters represent country name initials), NFHS—National Family Health Survey.

**Table 6 ijerph-18-01411-t006:** Other health outcomes that were based on DHS data on the association between cooking fuel use and different adverse health effect.

Authors and Publication Year	Country and Survey Year	Unit of Analysis	Sample Size	Exposed Group	Comparator Group	Outcome Definition	Covariate Adjusted (Table 1 Footnote)	Statistical Method and Main Results	Risk of BiasScore
Amegah et al. (2019)	Ghana, 2014 GDHS	women aged 15–49 years	4751	1. Charcoal, 2. Firewood and 3. Other biomass (straw/shrubs/grass or crop waste)	Clean fuel: electricity, LPG or natural gas	Reduced body weight and BMI	50, 40, 42, 30, 31, 43, 52 and 23	Linear regression Adjusted β (95% CI): Weight (in kg) Charcoal: −3.00 (−4.41, −1.60) Wood: −7.29 (−9.00, −5.58) Other biomass fuel: −4.10 (−7.15, 1.04) BMI (kg/m^2^) Charcoal: −0.78 (−1.50, −0.06) Wood: −2.27 (−2.95, −1.59) Other biomass fuel: −0.86 (−2.46, 0.75)	Moderate
Arku (2018)	Data collected between 2005 to 2014 in 10 countries	Premenopausal women (aged 15–49)	77,605	Solid fuels: coal, charcoal, wood, dung, crop waste, and shrub/grass	Clean fuels: electricity or gas	Average systolic BP ≥ 140 mmHg or average diastolic BP ≥ 90 mmHg and Hypertension	18, 30, 31, 43, 50, 52 and 78	Logistic regression Pooled OR (mmHg) (95% CI) Systolic BP: 0.58 (0.23, 0.93) Diastolic BP: 0.30 (−0.12, 0.72) Pulse pressure: 0.31 (−0.14, 0.75) Hypertension: 1.07 (0.99,1.16)	Moderate
Agrawal (2012)	India, 2005–2006 NFHS	People aged 20–60 years	99,574 women 56,742 men	Biomass fuel: wood, crop waste, dung, coal/lignite, charcoal or straw/shrubs/grass	Low-exposure group: kerosene, LPG, natural gas, biogas, or electricity	Asthma in women and men aged 20–49 years: Self-report, “Do you currently have Asthma?”	39, 40, 42, 30, 31, 57, 47, 45, 52, 50 and 51	Logistic regression OR, 95% CI: For women: 1.26 (1.06, 1.49) For men: 0.98 (0.77, 1.24)	Critical
Mishra (2003)	India, 1998–1999 NFHS	People with >60 years old	38,595	1. Only biomass fuels: wood, crop waste, or dung 2. Mixed fuel: biomass fuels plus cleaner fuels (coal/lignite or charcoal)	Cleaner fuels: kerosene, LPG, biogas, or electricity	Asthma in persons 60 or more years old: “Does anyone listed suffer from asthma?” Yes/No	39, 40, 41, 42, 43, 40, 31, 57, 47, 52, 50 and 51	Logistic regression OR, 95% CI: Only biomass: 1.59 (1.30, 1.94) Mixed fuel: 1.24 (1.04, 1.49)	Critical
Mishra (1999)	India, 1992–1993 NFHS	Person aged 20 years and above	260,162	Biomass fuels: wood or dung	Cleaner fuels: electricity, LPG, or biogas coal, charcoal or kerosene	TB in person aged 20 years and above: “Does anyone listed suffer from tuberculosis?” Yes/No	47, 57, 45, 40, 30, 31, 50 and 51	Logistic regression OR, 95% CI: 2.58 (1.98, 3.37)	Critical

LPG—liquefied petroleum gas, BP—blood pressure, BMI—body mass index, TB—tuberculosis, OR—odds ratio, DHS—Demographic and Health Survey (the first one or two letters represent country name initials), NFHS—National Family Health Survey.

**Table 7 ijerph-18-01411-t007:** Current challenges and suggested future directions for DHS data used in HAP and health studies.

Challenges	Suggestions (Survey Planning and Implementation/Policy/Research)	Priority to Implement	Difficulty to Implement
Data on the degree of exposure to smoke from cooking fuels collected using questionnaire, which is likely to be less accurate than other methods of exposure assessment.	DHS and other household surveys could implement more robust methods (e.g., biological monitoring, personal sampling and micro-environmental area-based sampling) in a smaller, representative sub-sample.	High	Medium
DHS collects a single (main) fuel item used for cooking, but households use multiple fuel items (fuel stacking) for cooking, and for other purposes (e.g., heating and lighting).	Modifying the DHS questionnaire in a way that it can capture multiple responses on the types of fuel used for cooking and for other purposes would be beneficial, including more granular estimates of exposure to cooking-derived pollutant (either categorical or continuous, from models and/or measurements).	High	Low
Included studies did not control for some potentially important ambient air pollutants (e.g., particulate matter, NO_2_, O_3_ and CO).	Existing public domain, global spatial exposure datasets (e.g., those used in the GBD and other studies) could be linked to geocoded DHS data, which would reduce the need for individual researchers to seek out the data and complete this linkage themselves.	Medium	Low
Most of the outcomes, except child nutritional status and a few birth weight outcomes, were based on self-report and prone to recall bias or have unclear validity.	Though it is very challenging, and unlikely to be achieved in the near future, a shift to collecting more objectively determined outcomes, either measured directly or through linkage with non-DHS administrative data, would enable a larger range and higher quality of analyses.	High	High
The included studies collectively suggest that missing data on cooking fuel and associated information (e.g., location of the kitchen and stove ventilation) can be make analyses problematic.	Explore implications of missing exposure data on health analyses and compare the utility of different imputation or prediction-based methods to deal with missingness.	High	Low

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
