# Peer review of "A Systematic Review and Appraisal of Epidemiological Studies on Household Fuel Use and Its Health Effects Using Demographic and Health Surveys"

_ijerph, 2021, doi:10.3390/ijerph18041411_

Round 1
Reviewer 1 Report
Overall: Thank you for the opportunity to review the manuscript entitled “A systematic review and appraisal of epidemiological studies on household fuel use and its health effects using the demographic and health surveys”. This study assessed the risk of bias and the quality of studies the investigate the health impacts of HAP. The assessment of quality and potential bias in research is a necessary and valuable task, and therefore the aim of this manuscript in important and will contribute to this field of research. However, I have a few comments or suggested changes, outlined below:
Specific comments:
Page 4: line 149 – This novel instrument is a key part of the methodology of this study and I found the description of this instrument quite vague or unclear. I understand there is a reference provided, however I highly suggest reviewing this section, and providing a clearer description of how you determined the quality and risk of bias for these studies.
Page 4: line 174 – once? Not one
Page 6: line 201 – Only 63 from 2785. What makes these studies irrelevant? Is this based on the exclusion criteria? Wrong primary outcomes?
Page 7: Line 249 – I would suggest you define what you mean by classification of exposures. Later in the discussion, you mention that some studies included HAP that were classed as low polluting which classed as a HAP after publication. Please include additional information.
Also, there are a few grammatical and errors in punctuation. Please proofread.
Author Response
Dear Editor,
Thank you for your correspondence on January 15 inviting us to submit a revised version of our manuscript. We appreciate the constructive comments of the three reviewers, and we have now responded to all of their specific comments and suggestions (please see below). We feel the revised manuscript has been strengthened as a result of the revisions.
Thank you once again for your consideration of our manuscript.
Sincerely,
Daniel B. Odo
Ian A. Yang
Luke D. Knibbs
Response to reviewers’ comments
We thank the three reviewers for their time and for their positive and constructive comments on our draft.
Reviewer 1
Comments and Suggestions for Authors
Overall: Thank you for the opportunity to review the manuscript entitled “A systematic review and appraisal of epidemiological studies on household fuel use and its health effects using the demographic and health surveys”. This study assessed the risk of bias and the quality of studies that investigate the health impacts of HAP. The assessment of quality and potential bias in research is a necessary and valuable task, and therefore the aim of this manuscript in important and will contribute to this field of research. However, I have a few comments or suggested changes, outlined below:
Specific comments:
Comment #1
Page 4: line 149 – This novel instrument is a key part of the methodology of this study and I found the description of this instrument quite vague or unclear. I understand there is a reference provided, however I highly suggest reviewing this section, and providing a clearer description of how you determined the quality and risk of bias for these studies.
Response
Thank you. We have added some new text to this section, and now it reads as follows (new text in bold):
The quality and risk of bias (RoB) of studies meeting the inclusion criteria were assessed, using a new RoB instrument for non-randomized studies (NRS) of exposures, [22] by one of us (D.B.O.). This tool seeks to assess the methodological quality of the evidence and the RoB of studies of environmental exposures. Guidance on the application of the tool is detailed elsewhere [22], including how to interpret and present the RoB of the studies, and using that information to make a transparent judgment. The three steps involved in applying this instrument were: (1) present the review question, potential confounders, and exposure and outcome measurement accuracy information, (2) describe each eligible study as a hypothetical target experiment and including specific con-founders from that study that will require consideration, and (3) assess RoB across seven items on the strengths and limitations of studies. As a general indicator, a ‘low’ RoB study would control important confound-ers listed in the RoB assessment table, would ascertain health outcome objectively (with validated method) and supported with record linkage (Table S2).
Comment #2
Page 4: line 174 – once? Not one
Response
Thank you. We have modified our wording and now it reads as follows (new text in bold):
We conducted RoB assessment at the outcome level. Because RoB can vary within a study if multiple outcomes are assessed, we therefore evaluated RoB for each outcome.
Comment #3
Page 6: line 201 – Only 63 from 2785. What makes these studies irrelevant? Is this based on the exclusion criteria? Wrong primary outcomes?
Response
Based on this comment, we have clarified this point in our PRISMA flow chart (Figure 1), as well in the methods section where we included the descriptions why these records/studies were excluded. Now, as it can be seen in our flowchart, 2685 records were excluded because they were not an epidemiological study, and/or exposure variable not relevant to the topic, and/or not using DHS data, and/or had a health outcome that was undefined.
Comment #4
Page 7: Line 249 – I would suggest you define what you mean by classification of exposures. Later in the discussion, you mention that some studies included HAP that were classed as low polluting which classed as a HAP after publication. Please include additional information.
Response
Thank you. We clarified what classification of exposure mean in our study both in the results and in the discussion part, and now it reads as follows (new text in bold):
In the results section:
…Based on the item-level RoB assessment, three of the seven domains where all the studies included in this review scored low RoB were on the selection of participants, missing data and selection of reported results, whilst confounding and classification of exposure (i.e., separating households into those using relatively clean and relatively polluting fuels) were the two domains resulted in serious or critical RoB. Results of RoB assessment and grading for each study- and item-level presented in the supplementary file (Table S3).
And the discussion section:
…Although anthropometric information in the DHS surveys is collected objectively with standardized methods, exposure misclassification (mixing relatively clean and relatively polluting fuels in assigning households to exposed and reference groups) was the main reason studies of childhood stunting and anaemia were rated as having serious RoB.
Comment #5
Also, there are a few grammatical and errors in punctuation. Please proofread.
Response
Thank you. We have corrected grammatical and punctuation errors in the revised version.
Reviewer 2 Report
The research is well presented, the methodology is adequately developed an explained, the results, conclusions and discussion are adequately presented and the limitations of the study are clearly stated.
Both the introduction and the discussion can be improved with additional information to contextualize the work on a global context and regarding previously developed research.
Specifically, the explanation about the scope of the study could be improved since, for example, it does not consider any study in Latin America. There are various countries in Latin America where firewood is widely used as fuel for cooking and heating, which has important impacts on the health of the population. In these countries there are studies that analyze Demografic and Health Surveys and that should be considered in this work. They may be more difficult to find since in many cases the same equipment is used for cooking and heating.
The study should clarify whether the Systematic Review only focuses on some continents and countries. In addition, it is important to explain if only equipment used for cooking or also for cooking and heating are considered, the latter are very common in some countries.
In addition, in many of these countries efficient equipment and efficient wood-based fuels are used, which do not produce pollution inside the households but produce large amounts of emissions to the exterior, that ultimately ends up impacting people's health. Therefore it is important to comment on these differences or explain that these cases have been left out of the study.
Are only open fires considered or all wood-based cooking equipment? There are quite a few differences in this aspect among countries and it is necessary to clarify.
Author Response
Dear Editor,
Thank you for your correspondence on January 15 inviting us to submit a revised version of our manuscript. We appreciate the constructive comments of the three reviewers, and we have now responded to all of their specific comments and suggestions (please see below). We feel the revised manuscript has been strengthened as a result of the revisions.
Thank you once again for your consideration of our manuscript.
Sincerely,
Daniel B. Odo
Ian A. Yang
Luke D. Knibbs
Response to reviewers’ comments
We thank the three reviewers for their time and for their positive and constructive comments on our draft.
Reviewer 2
Comments and Suggestions for Authors
The research is well presented, the methodology is adequately developed an explained, the results, conclusions and discussion are adequately presented and the limitations of the study are clearly stated.
Comment #1
Both the introduction and the discussion can be improved with additional information to contextualize the work on a global context and regarding previously developed research.
Specifically, the explanation about the scope of the study could be improved since, for example, it does not consider any study in Latin America. There are various countries in Latin America where firewood is widely used as fuel for cooking and heating, which has important impacts on the health of the population. In these countries there are studies that analyze Demographic and Health Surveys and that should be considered in this work. They may be more difficult to find since in many cases the same equipment is used for cooking and heating.
Response
Thank you. We agree that the reach of the DHS into Latin America is important as solid fuel use prevalence can be high in some countries. However, our review did not identify any publications based on DHS data collected in that region that met our inclusion criteria. It is possible that there are articles published outside of the English language literature. The scope of our review was restricted to articles published in English because we did not have capacity to repeat our searches in multiple databases, spanning the diverse number of language groups spoken across DHS countries.
We have made some modifications to our previous wordings that explained the scope of our review both in the introduction and in the method section
This section reads as follows (new text in bold):
The introduction section:
…Here, we aimed to: (i) identify and collate all relevant peer-reviewed epidemiological studies of DHS-derived HAP estimates and health performed globally, (ii) determine what variables and analytical approaches can place studies at greater risk of bias, using a novel rating tool, and (iii) identify additional variables that could boost utility without being onerous to collect in resource-limited settings, given the main purpose is not to collect data for HAP research. Due to practical constraints, this review did not include articles published in languages other than English.
And the methods section:
…In brief, peer-reviewed studies that were based entirely on DHS data and considered HAP as a main exposure variable were included, with no restriction on the type of health outcome, geographic location (country), age or gender of study participants. All searches were restricted to articles published in English since the inception of the DHS program (1985) through to September, 2020.
In addition, we have now stated in the limitations section that we could have missed articles that were published in languages other than English.
This section reads as follows (new text in bold):
This review has several limitations. One of the limitation of our re-view is that we have included peer-reviewed research articles published only in English language journals. As the DHS surveys are also conducted in non-English speaking countries and the survey information is widely used to develop government documents (e.g., working papers), we might have missed grey literature and some records published in languages other than English. However, we do not think this would alter our substantive findings, as the data collected under the DHS program, and used by researchers, are similar. Finally, for practical reasons the RoB assessment was undertaken by one of us, and despite using a standard process for assessing each article’s RoB, errors or omissions may mean the RoB has been under- or over-estimated for individual studies.
Comment #2
The study should clarify whether the Systematic Review only focuses on some continents and countries.
Response
There was no restriction based on study location. Please see our response to comment #1 for further details.
Comment #3
In addition, it is important to explain if only equipment used for cooking or also for cooking and heating are considered, the latter are very common in some countries.
Response
We fully agree that polluting fuels can be used for different purposes in households such as cooking, heating and lighting. Our review focused on epidemiological evidence that has analysed from the Demographic and Health Survey (DHS) surveys, and these surveys only collect information on the main fuel type used for cooking. Indeed, as we described in the discussion section of the previous and current versions (page 1: lines 446–449), the fact that the surveys do not collect information on fuels for heating or lighting is one of the key limitations of the DHS surveys for epidemiological analyses.
Comment #4
In addition, in many of these countries efficient equipment and efficient wood-based fuels are used, which do not produce pollution inside the households but produce large amounts of emissions to the exterior, that ultimately ends up impacting people's health. Therefore it is important to comment on these differences or explain that these cases have been left out of the study.
Response
Thank you. We agree that different stoves have different emission levels; some of them are more efficient than others but all affect outdoor air quality. We have now added to the discussion section under ‘Limitations of DHS Surveys (With Respect to HAP and Health)’, and it reads as follows:
….Third, different stoves using the same fuel can vary markedly in their emission characteristics [1], but this level of detail is not collected in the DHS. Cooking emissions also affect outdoor air quality, which is beyond the DHS’ remit, but presents an important challenge for researchers attempting to select covariates in their analyses of cooking fuel and health associations.
Comment #5
Are only open fires considered or all wood-based cooking equipment? There are quite a few differences in this aspect among countries and it is necessary to clarify.
Response
We agree that there are several differences in wood stoves around the world. As we state in our response to comment #4, the DHS does not capture such subtleties, which is an important limitation of it.
Reviewer 3 Report
Thank you very much for the opportunity to critique this review paper that characterizes household fuel usage and associated health effects in many low and middle income countries around the world. The authors - Odo et al., use the Demographic and Health Surveys to obtain the requisite data and conduct the overall analyses and review.
At the outset, it is obvious that authors have put in a lot of effort in this systematic review. I have perused in detail all the supplementary documents as well and I can attest that this systematic review is a byproduct of dedication and hard work - especially by the primary author.
The authors have also incorporated analyses on the statistical bias, confounders etc. for the studies they reviewed. I would recommend the publication of this manuscript but I have some reservations and comments and would appreciate the authors' input before this paper can be finally published. My comments are as follows:
Lines 172- 175: This is a minor comment. Please rewrite this part again. It is just too wordy and the reader misses the point. I am sure there is a better way of discussing risk of bias assessment at the outcome level.
Lines 206- 228: This is the major and most important comment that I have for the authors. When one reads through the manuscript, it becomes obvious that the countries included in the analyses are all Anglophone Nations or countries where English is one of the official languages or were former British Colonies and are now part of the British Commonwealth. Case in point - the nations in the Indian Sub-Continent, East African nations like Kenya, Uganda, Tanzania. Now when we discuss air pollution and household fuel usage, how can we forget the Sub-Saharan African Francophone Nations such as DRC, Togo, Benin, Burkina Faso, Senegal, Cote D'Ivoire, Mali, Niger, Chad, Guinea, Rwanda, Burundi, Republic of Congo, Central African Republic etc. !!! The only Francophone country from Sub-Saharan Africa included is Gabon. There is no information or analyses for all these low and middle income nations. Furthermore, let us shift focus now to Central American Nations (Guatemala, El Salvador, Nicaragua, Panama, Honduras) and Latin American Nations (Ecuador, Peru, Bolivia, and many others...) for which the authors have provided no information or data analyses. All these countries mentioned in the Spanish speaking belt of the Americas also face severe air pollution problems because of household fuel usage. In fact, a lot of researchers have done research on indoor air pollution in Guatemala and other nations and there is a rich treasure trove of published literature from these nations. The systematic review presented here has not considered these countries !
This reviewer understands that the authors may not be French or Spanish speakers but retrieving information about these countries is crucial for this analyses. If the authors decide to overlook these countries, then perhaps they can change the title of the manuscript and indicate that these reviews pertain to nations where English is one of the official languages.
Lines 549-562: The authors have put in a lot of effort in this review as I have mentioned above. The conclusion needs to be expanded a bit more. Discuss the importance of the review conducted and its contribution to the overall body of literature versus just focusing on the limitations etc.
Author Response
Dear Editor,
Thank you for your correspondence on January 15 inviting us to submit a revised version of our manuscript. We appreciate the constructive comments of the three reviewers, and we have now responded to all of their specific comments and suggestions (please see below). We feel the revised manuscript has been strengthened as a result of the revisions.
Thank you once again for your consideration of our manuscript.
Sincerely,
Daniel B. Odo
Ian A. Yang
Luke D. Knibbs
Response to reviewers’ comments
We thank the three reviewers for their time and for their positive and constructive comments on our draft.
Reviewer 3
Thank you very much for the opportunity to critique this review paper that characterizes household fuel usage and associated health effects in many low and middle income countries around the world. The authors - Odo et al., use the Demographic and Health Surveys to obtain the requisite data and conduct the overall analyses and review.
At the outset, it is obvious that authors have put in a lot of effort in this systematic review. I have perused in detail all the supplementary documents as well and I can attest that this systematic review is a byproduct of dedication and hard work - especially by the primary author.
The authors have also incorporated analyses on the statistical bias, confounders etc. for the studies they reviewed. I would recommend the publication of this manuscript but I have some reservations and comments and would appreciate the authors' input before this paper can be finally published. My comments are as follows:
Comment #1
Lines 172- 175: This is a minor comment. Please rewrite this part again. It is just too wordy and the reader misses the point. I am sure there is a better way of discussing risk of bias assessment at the outcome level.
Response
Thank you. We have modified our wording and now it reads as follows (new text in bold):
We conducted RoB assessment at the outcome level. Because RoB can vary within a study if multiple outcomes are assessed, we therefore evaluated RoB for each outcome.
Comment #2
Lines 206- 228: This is the major and most important comment that I have for the authors. When one reads through the manuscript, it becomes obvious that the countries included in the analyses are all Anglophone Nations or countries where English is one of the official languages or were former British Colonies and are now part of the British Commonwealth. Case in point - the nations in the Indian Sub-Continent, East African nations like Kenya, Uganda, Tanzania. Now when we discuss air pollution and household fuel usage, how can we forget the Sub-Saharan African Francophone Nations such as DRC, Togo, Benin, Burkina Faso, Senegal, Cote D'Ivoire, Mali, Niger, Chad, Guinea, Rwanda, Burundi, Republic of Congo, Central African Republic etc. !!! The only Francophone country from Sub-Saharan Africa included is Gabon. There is no information or analyses for all these low and middle income nations. Furthermore, let us shift focus now to Central American Nations (Guatemala, El Salvador, Nicaragua, Panama, Honduras) and Latin American Nations (Ecuador, Peru, Bolivia, and many others...) for which the authors have provided no information or data analyses. All these countries mentioned in the Spanish speaking belt of the Americas also face severe air pollution problems because of household fuel usage. In fact, a lot of researchers have done research on indoor air pollution in Guatemala and other nations and there is a rich treasure trove of published literature from these nations. The systematic review presented here has not considered these countries!
This reviewer understands that the authors may not be French or Spanish speakers but retrieving information about these countries is crucial for this analyses. If the authors decide to overlook these countries, then perhaps they can change the title of the manuscript and indicate that these reviews pertain to nations where English is one of the official languages.
Response
Thank you. We agree that the reach of the DHS into countries where their official language is different from English is important as solid fuel use prevalence can be high in some of those countries. However, our review did not identify any publications based on DHS data collected in that region that met our inclusion criteria. It is possible that there are articles published outside of the English language literature. The scope of our review was restricted to articles published in English because we did not have capacity to repeat our searches in multiple databases, spanning the diverse number of language groups spoken across DHS countries.
We have made some modifications to our previous wordings that explained the scope of our review in the introduction.
This section reads as follows (new text in bold):
The introduction section:
Here, we aimed to: (i) identify and collate all relevant peer-reviewed epidemiological studies of DHS-derived HAP estimates and health performed globally, (ii) determine what variables and analytical approaches can place studies at greater risk of bias, using a novel rating tool, and (iii) identify additional variables that could boost utility without being onerous to collect in resource-limited settings, given the main purpose is not to collect data for HAP research. Due to practical constraints, this review did not include articles published in languages other than English.
In addition, we have now stated in the limitations section that we could have missed articles that were published in languages other than English
This section reads as follows (new text in bold):
This review has several limitations. One of the limitation of our re-view is that we have included peer-reviewed research articles published only in English language journals. As the DHS surveys are also conducted in non-English speaking countries and the survey information is widely used to develop government documents (e.g., working papers), we might have missed grey literature and some records published in languages other than English. However, we do not think this would alter our substantive findings, as the data collected under the DHS program, and used by researchers, are similar. Finally, for practical reasons the RoB assessment was undertaken by one of us, and despite using a standard process for assessing each article’s RoB, errors or omissions may mean the RoB has been under- or over-estimated for individual studies.
Comment #3
Lines 549-562: The authors have put in a lot of effort in this review as I have mentioned above. The conclusion needs to be expanded a bit more. Discuss the importance of the review conducted and its contribution to the overall body of literature versus just focusing on the limitations etc.
Response
Thank you. We have added some additional text to the conclusion, which now reads as follows (new text in bold):
Although HAP exposure assessment is not the main focus of the DHS, it is the main, often only, source of information in many low- and middle-income countries. It remains an important resource, and researchers have used it to perform important cross-sectional analyses on HAP from cooking fuels and health in 57 countries. By restricting our focus to DHS-based studies, our review allowed a greater granularity in our assessment of DHS-specific issues. Through that, we found that an appreciable proportion of the studies we reviewed have potential for serious or critical RoB, due mainly to the inherent limitations of what information can feasibly collected in such a large-scale survey, and methodological choices by end-users. This relates to confounder control, exposure misclassification, and outcome ascertainment. We underscore to researchers the need to bear these issues in mind in their analyses. Moreover, we also offer some pragmatic suggestions through which DHS data could be even more amenable to studies of household fuel use and health, and reduce the RoB, without being onerous to collect or analyse. We hope our review is useful for researchers and others working with DHS data.